# Development of a Rapid Fluorescent Diagnostic System for Early Detection of the Highly Pathogenic Avian Influenza H5 Clade 2.3.4.4 Viruses in Chicken Stool

**DOI:** 10.3390/ijms23116301

**Published:** 2022-06-04

**Authors:** Bao-Tuan Duong, Duc-Duong Than, Bae-Gum Ju, Thuy-Tien Thi Trinh, Chris-Ka Pun Mok, Ju-Hwan Jeong, Min-Suk Song, Yun-Hee Baek, Hyun Park, Seon-Ju Yeo

**Affiliations:** 1Zoonosis Research Center, Department of Infection Biology, School of Medicine, Wonkwang University, Iksan 54538, Korea; bao2dt@gmail.com (B.-T.D.); ducduong27189@gmail.com (D.-D.T.); bgj4814@gmail.com (B.-G.J.); 2Department of Tropical Medicine and Parasitology, Medical Research Center, Institute of Endemic Diseases, Seoul National University, Seoul 03080, Korea; thuytientrinh@snu.ac.kr; 3The Jockey Club School of Public Health and Primary Care, The Chinese University of Hong Kong, Shatin, Hong Kong 999077, China; kapunmok@cuhk.edu.hk; 4Li Ka Shing Institute of Health Sciences, Faculty of Medicine, The Chinese University of Hong Kong, Hong Kong 999077, China; 5Department of Microbiology, College of Medicine and Medical Research Institute, Chungbuk National University, Cheongju 28644, Korea; jeongbau07@naver.com (J.-H.J.); songminsuk@chungbuk.ac.kr (M.-S.S.); microuni@chungbuk.ac.kr (Y.-H.B.); 6Department of Tropical Medicine and Parasitology, Department of Biomedical Sciences, College of Medicine, Seoul National University, Seoul 03080, Korea

**Keywords:** highly pathogenic avian influenza A(H5Nx), human influenza virus infections, monoclonal antibody, rapid fluorescent immunochromatographic strip test

## Abstract

Rapid diagnosis is essential for the control and prevention of H5 highly pathogenic avian influenza viruses (HPAIVs). However, highly sensitive and rapid diagnostic systems have shown limited performance due to specific antibody scarcity. In this study, two novel specific monoclonal antibodies (mAbs) for clade 2.3.4.4 H5Nx viruses were developed by using an immunogen from a reversed genetic influenza virus (RGV). These mAbs were combined with fluorescence europium nanoparticles and an optimized lysis buffer, which were further used for developing a fluorescent immunochromatographic rapid strip test (FICT) for early detection of H5Nx influenza viruses on chicken stool samples. The result indicates that the limit of detection (LoD) of the developed FICT was 40 HAU/mL for detection of HPAIV H5 clade 2.3.4.4b in spiked chicken stool samples, which corresponded to 4.78 × 10^4^ RNA copies as obtained from real-time polymerase chain reaction (RT-PCR). An experimental challenge of chicken with H5N6 HPAIV is lethal for chicken three days post-infection (DPI). Interestingly, our FICT could detect H5N6 in stool samples at 2 DPI earlier, with 100% relative sensitivity in comparison with RT-PCR, and it showed 50% higher sensitivity than the traditional colloidal gold-based rapid diagnostic test using the same mAbs pair. In conclusion, our rapid diagnostic method can be utilized for the early detection of H5Nx 2.3.4.4 HPAIVs in avian fecal samples from poultry farms or for influenza surveillance in wild migratory birds.

## 1. Introduction

Highly pathogenic avian influenza viruses (HPAIVs) of subtype H5 were first isolated in China in 1996, and classified into ten clades and many subclades in the following two decades [1]. H5 HPAIVs cause disease and high mortality in both avians and humans, while H5 low pathogenicity avian influenza viruses (LPAIVs) typically cause little or no clinical signs in infected birds; however, they have the potential to change to HPAIVs through mutation [2]. Within H5 HPAIVs, clade 2.3.4.4 was first reported in China in 2013, but the global circulation of these viruses led to further classification into eight subclades (2.3.4.4a–h) [3]. These subclades were responsible for avian influenza outbreaks in Korea and Japan in early 2014 [4,5]. Until October 2018, subclades 2.3.4. 4a–h had been detected in 51 countries, including Germany, The Netherlands, the United Kingdom, Italy, the Russian Federation, and the United States, with the epidemic in commercial poultry resulting in the death or culling of approximately 43 million chickens and 7 million turkeys [5,6]. During 2021, seven cases of H5N8 and five case of H5N6/2.3.4.4b human infections were reported in the Russian Federation and China, respectively [7,8]. To date, 42 cases of human infection with influenza A(H5N6) 2.3.4.4 viruses have been reported, 22 of which resulting in death [7,9,10]. Moreover, antibodies against 2.3.4.4 viruses were found in the sera of many residents during the 2017−2018 H5 HPAI outbreak in Vietnam [11]. Taken together, these results strongly suggest that viruses causing poultry outbreaks can also cause human H5 HPAI cases. Thus, diagnostic systems targeting 2.3.4.4 subclades of HPAI H5 viruses could prevent the spreading of HPAI as a primary screening clinical tool. However, the development of such techniques has been hindered by many factors including the lack of antigen expression systems for producing monoclonal antibodies (mAbs) [12].

*Escherichia coli* is the most common host for producing large amounts of recombinant proteins with high expression levels [13]. Unfortunately, such proteins lack glycosylation sites and the necessary folding to maintain proper biological function [13,14]. To solve this problem, researchers have been replacing *E. coli* with other expression hosts for producing immunogenic antigens, as well as new vaccine candidates [15,16,17]. Despite these efforts and various reports of HPAI H5-specific antibodies, HPAI H5-specific rapid diagnostic tests (RDTs) performance on fecal samples is still scarce [18,19,20]. Many research groups have focused on the development of such RDTs by targeting the hemagglutinin (HA) of H5 viruses [20,21]. Nevertheless, the sensitivity of these RDTs is lower than that of RDTs targeting conserved regions, and they are unable to discriminate between H5 HPAIVs and H5 LPAIVs [20]. Additionally, RDT kits are commonly used to detect viruses in the trachea and cloaca, or in tissue homogenates of clinical samples, lacking sensitivity for detecting viruses in fecal samples, as the high density, mucus, color, or pH of feces can interfere with RDT reactions [18,20]. However, fecal samples are the most convenient and suitable to collect from poultry and wild birds in their habitats without any host contact, and chicken feces are known to contain the highest viral load [22]. Therefore, high-performance H5 HPAIVs RDTs for fecal samples will be key for the identification and surveillance of influenza in poultry, wild birds, and humans.

In the present study, we aimed to develop a rapid fluorescent diagnostic system for early detection of the H5 HPAIVs subclade 2.3.4.4, one of the most emerging variant subclades of the H5 HPAIVs lineage, in order to distinguish it from other H5 LPAIVs in chicken feces [23,24]. The *E. coli* host was replaced by reverse genetics of influenza viruses based on a mammalian expression system to develop H5Nx 2.3.4.4-specific mAbs, which were combined with a fluorescence material and suitable lysis buffer to discriminate between H5 HPAIVs and H5 LPAIVs. A chicken model was also used to evaluate the ability of these mAbs for early and specific detection of H5 HPAIV 2.3.4.4 subclades on fecal samples.

## 2. Results

### 2.1. Generation of Recombinant Virus by Reverse Genetics and Antigen Expression in E. coli 

*E. coli* expression systems have been reported to show low performance and to produce deficient functional proteins that may be affected by the absence of glycosylation and folding defects [25]. In the present study, the performance of the mammalian reverse genetics expression system was compared to that of the *E. coli* to evaluate the optimum host for developing H5Nx-specific mAbs. As such, we obtained H5 RGV by cloning seven PR8/H1N1 genes and the H5N6 2.3.4.4b HA gene segment into the pHW2000 expression vector, which was then transfected into a mixture of MDCK and 293T cells. The H5 RGV showed cytopathic effects three days post-transfection (DPT) and the virus titer was at 1280 HAU/mL at five DPT according to the hemagglutination assay (Figure 1A and Appendix A). The presence of H5 RGV was confirmed by PCR using H5-specific primers as a specific band of 249 bp was produced (Appendix A). We also cloned the globular head domain sequence of the same H5N6 HA segment into the pET21b vector for expression in the *E.coli* host. This domain is the most important antigen-binding site of influenza A viruses (Appendix A) [26]. As the antigen was expressed in insoluble form after 0.5 mM isopropyl-b-D-thiogalactopyranoside induction, it was solubilized under denaturing conditions and then refolded for purification to obtain the pure antigen with ~30 kDa (Figure 1B). Both H5 RGV and H5N6 HA antigen were mixed with an adjuvant before mice immunization.

### 2.2. Production and Screening of Mice mAbs

Footpad injection is a safe and effective method of mouse immunization to produce specific antibodies [27,28]. In this experiment, we immunized BALB/c female mice with alive H5 RGV or H5N6 HA recombinant antigen produced by *E. coli* using the footpad injection method. Two weeks after each immunization, mice sera were sampled for checking antibody levels against H5 RGV by indirect ELISA. As shown in Figure 2A, antibody level was dramatically increased after the second and third immunizations with H5 RGV in comparison with immunizations with H5N6 HA recombinant antigen from *E. coli* (***, *p* < 0.001). Forty-two days post-injection, lymphoid node and spleen tissues were collected for cell fusion using the hybridoma technology (Appendix A). Twelve candidate antibodies were obtained as primary hybridoma clone candidates for detection of H5 RGV in both immunization groups (Figure 2B,C). Among them, three primary candidates (#11, #23, and #9) were selected as potential H5-specific mAbs. Using the limiting dilution method, mAbs #23.3 and #11.4 (H5 RGV immunogen group) and mAb #9.1 (H5N6 HA *E. coli* host immunogen group) were obtained for specific detection of H5 viruses (Figure 2D). Finally, #23.3, #11.4, and #9.1 were selected for further characterization.

### 2.3. Characterization of mAbs

This experiment aimed to characterize the potential H5-specific mAbs. We first obtained three mAb candidates from the fluid of mice ascites using protein G columns. The results in Figure 3A,B indicate their specificity for detecting H5 virus subtypes. Interestingly, the #23.3 and #11.4 mAbs derived from the H5 RGV immunogen group could broadly detect H5 RGV, H5N8 RGV, and H5N3 LPAI virus subtypes, while #9.1 derived from the HA H5N6 *E. coli* expression host could only detect H5 RGV. These data indicate that the antibodies derived from the RGV expression system were able to broadly and highly detect H5 subtype viruses compared to that derived from the *E. coli* expression host. Furthermore, the KD values of three mAbs were measured by surface plasmon resonance to determine their affinity to the H5N6 HA antigen expressed in the *E. coli* host. The binding affinity of antibodies was low and within the nanomolar range of 10^−8^ to 10^−10^ (#11.4, KD = 5.864 × 10^−8^; #23.3, KD = 7.572 × 10^−10^; #9.1; KD = 1.242 × 10^−10^ M) (Appendix A). These KD values were comparable among the mAbs derived from both expression systems and revealed strong affinity to the H5N6 HA antigen. However, antibodies derived from the H5 RGV system were more likely to bind to real viruses than the antibody derived from the *E. coli* host (Figure 2D).

### 2.4. Screening of the Specific mAb Pair for FICT

Many assays are available to select antibody pairs for application on RDT strips [29,30]. In the present study, mAb pairs were determined via sandwich FLISA (Figure 4). The results showed that the #11.4/detector and the #23.3/coating mAb pair significantly enhanced the fluorescence signal for detection of both H5/2.3.4.4b RGV and H5/2.3.4.4c RGV HPAIVs (*** *p* < 0.001) (Figure 4A). These antibodies were then selected for FICT (Appendix A). 

### 2.5. Lysis Buffer Optimization

Using this potential mAb pair, we then investigated suitable conditions for virus detection by FICT. Indeed, pH and detergents have been reported as important for removing nonspecific binding and increasing the reactivity of mAbs in immunoassays [29,31]. In the present study, nonspecific reactions and the binding capability of the bio-conjugate was investigated by using different pH (6, 9, and 11) and SDS at 0.3 to 0.9% in the lysis buffer. At pH 6 and 0.6 or 0.9% SDS, the bio-conjugate aggregated in the nitrocellulose membrane, resulting in the low performance of virus detection (Appendix A). Interestingly, the lysis buffer could remove nonspecific reactions and maintain the ratio of TL/CL between the virus-infected and the control groups when the pH range increased to 9.0 and 11 (Figure 5B–F). Overall, the lysis buffer at pH 9.0 containing 0.6% SDS showed the best performance, with bio-conjugate smooth migration without any nonspecific binding on the nitrocellulose membrane and the highest ratio of TL/CL (Figure 5B). These results indicated that the lysis buffer played a key role in the reaction of bio-conjugates, with a suitable amount of detergent and pH helping to expose targeted epitopes from other proteins to enhance antibody and antigen reaction.

### 2.6. Specificity and LoD of FICT

We then tested the LoD of FICT using virus-spiked samples to demonstrate the ability of this assay for H5 HPAIVs detection. As shown in Figure 6A, the LoD of our system was 5 HAU/mL for H5/2.3.4.4b RGV detection, while the LoD of the colloidal gold-based RDT kit was 160 HAU/mL. The results indicated that the Eu NP-conjugated antibody-based FICT increased virus detection capacity by 32-fold compared to that done by the colloidal gold-based RDT (Figure 6C). We then assessed the ability of FICT for detecting H5/2.3.4.4b RGV in the mimic clinical (spiked) samples. H5/2.3.4.4b RGV was spiked in chicken stool or human nasal mucus at the different virus titers, and the LoD for the mimic samples was determined by FICT. As shown in Figure 6B,D, the LoD was 40 HAU/mL and 20 HAU/mL in chicken stool and human nasal mucus spiked samples, respectively. Interestingly, the intensity signals decreased when increasing amounts of chicken stool were used for FICT testing, but the LoD remained 40 HAU/mL with good linear regression fit (R^2^ > 0.9). These results revealed that our FITC assay was stable across different amounts of fecal samples and could detect the virus in all samples, which is crucial owing to the influence of feces amount on antigen-antibody reactions.

Next, we evaluated FICT performance by comparing its LoD with that of RT-PCR. The RNA corresponding to the LoD point of each sample was extracted and then used for RT-PCR. The plot of the standard curve of Ct values against the logarithmic dilutions produced a R^2^ value of 0.991 and the slope corresponded to an 89.9% efficiency, which was close to an optimized protocol. The LoDs of 5 HAU/mL (DW), 40 HAU/mL (chicken stool), and 20 HAU/mL (human nasal swab) for H5/2.3.4.4b RGV-spiked samples corresponded to Ct values of 27.34, 27.55, and 25.35, respectively, and FICT RNA copy numbers of 5.53 × 10^4^ ± 1.01 × 10^4^, 4.784 × 10^4^ ± 3.063 × 10^4^, and 1.955 × 10^5^ ± 1.553 × 10^4^, respectively (Figure 7). These were within the range of virus RNA copy number in the specimen sample [32]. Therefore, our FICT assay using fluorescent Eu could detect the H5/2.3.4.4b virus in specimen samples.

We then evaluated the specificity of FICT by testing other virus subtypes, including H1N1, H2N9, H3N2, H9N2, H5N3 LPAI, H5/2.3.4.4b RGV, H5/2.3.4.4c RGV HPAI, and Influenza B viruses at high titer (640 HAU/mL). These subtypes were also subjected to the SD RDT commercial kit assay for the normalization of virus amount between the two systems. As shown in Figure 8A, the FICT assay detected only H5 HPAIVs (***, *p* < 0.001) and could not recognize the other subtypes, including H5N3 LPAIV. Contrarily, all virus subtypes were positive in the RDT commercial kit (Figure 8B). Therefore, our FICT assay had H5 HPAIVs specificity. 

### 2.7. Evaluation of FICT System in the Chicken Model

We then demonstrated the performance of our FICT system for H5N6 virus detection in feces samples in a clinical study. Four-week-old specific-pathogen-free chicken were infected with A(H5N6)/Duck/Foshan/41/2019 or H9N2 virus, or non-inoculated (n = 3 in each group). Fecal and cloacal samples were collected two days post-infection (DPI). During the incubation period of viruses, A(H5N6)/Duck/Foshan/41/2019-infected chicken showed severe clinical signs at 2 DPI and 100% chicken lethality at 3 DPI, while clinical signs were not observed in the A/Chicken/Korea/LPM429/2016 (H9N2) infected group. The matrix protein target was selected for virus quantification by RT-PCR, and positive samples were established at Ct ≤ 37 with low Ct values, therefore indicating high viral load in the tested samples [33]. In the present study, positive Ct values were observed at 2 DPI in fecal and cloacal samples of both H5N6- and H9N2-infected groups. The Ct value range of H5N6 in fecal samples (25.89 to 29.31) was lower than that in cloacal samples (27.52 to 30.51) and a similar trend was found for H9N2 Ct values (23.38 to 25.30 in fecal samples and 29.61 to 33.92 in cloacal samples) (Appendix A). These Ct values indicated a higher viral load of both H5N6 and H9N2 in fecal samples than in cloacal samples.

The cut-off value of FICT was based on the highest TL/CL ratio in the non-inoculated (control) group; cut-off > 1.56; 0.3 for positive samples in fecal and cloacal samples, respectively (Appendix A). As shown in Table 1 and Figure 9, our FICT system had high sensitivity for the early detection of H5N6 in stool samples (at 2 dpi 100% (4/4) detection), while no positive result was detected in cloacal samples. In addition, there was non-specific binding to H9N2 or in the control group. As such, FICT showed 100% sensitivity in comparison with RT-PCR for H5N6 detection in chicken feces. The colloidal gold-based RDT kit using the same pair of mAbs could detect two of the four positives in feces samples, thus showing 50% sensitivity (Appendix A). Overall, our FICT system was optimized for the early and specific detection of H5N6 in fecal samples, showing higher sensitivity than the traditional colloidal gold-based RDT.

## 3. Discussion

Given the ongoing COVID-19 pandemic and the constant emergence of novel variants of pathogens of concern, including H5Nx HPAIVs, a diagnosis system for the early detection of viruses and their related materials, including those used for antigen and antibody development, is of utmost importance. Instead of using the *E. coli* host expression system, many research groups have used a mammalian host system for antigen expression to produce a specific antibody [29,34]. Mammalian host-synthesized proteins have complex folding and post-translational modifications, such as glycosylation [35]. However, these systems are protein yield-limited and costly [36]. To solve this problem, we used a RGV system based on mammalian expression systems for rapidly reconstructing a high titer recombinant virus to produce specific mAbs. RGV is a well-known and powerful research tool for developing vaccines, as well as for studying the effects of mutations on the functions of viral genes and their interactions with the hosts [37]. Recently, RGV and the virus itself have been used for producing specific mAbs [20,28,38]. According to the World Health Organisation, HPAIVs should be handled at facilities with BSL 3 or higher, which is limiting, costly, and increases the difficulty of performing experiments [39,40]. However, modification of HA cleavage sites by RGV allows working with HPAIVs at BSL 2 facilities at low cost [41]. Additionally, virus preparation requires less time (<1 week) without any purification steps, and a very low virus titer (128 HAU) for mice immunization. Although *E. coli* host expression systems are common for accumulating proteins in inclusion bodies, it usually requires two weeks for antigen preparation, refolding, and purification before mice immunization [42,43]. Furthermore, their immunogenicity and derived antibodies are less efficient in detecting H5 HPAIs viruses, as demonstrated in the present study. Taken together, our results revealed that reverse genetics of influenza virus was a more advantageous technique for developing H5Nx-specific mAbs than the *E. coli* expression host.

The H5Nx HPAIV 2.3.4.4 clade has been further classified into eight subclades (2.3.4.4a–h) based on their antigenicity diversity [41]. Among them, subclades 2.3.4.4b and h are the most dominant in terms of global circulation and human infection cases [7,8,44]. These subclades are antigenically distinct from each other, as well as from the viruses in other 2.3.4.4 subclades, which reduces the sensitivity and specificity of cross-antibodies against H5 HPAI 2.3.4.4b and h subclades [20,45]. Therefore, the generation of novel antibodies should be considered for the development of RDT kits for these two subclades (Figure 10).

Multiple research groups have been focused on the development of mAbs for detecting H5 influenza viruses, and the LoD of H5-specific RDTs currently ranges from 0.23 ng/mL to 2.8 μg/mL for HA antigen detection [19,46]. The present study could not confirm or validate the detection of H5 HPAIVs within the 2.3.4.4 clade using commercial RDTs. To the best of our knowledge, only one study has reported the rapid and specific detection of H5 HPAIVs 2.3.4.4 subclades [20]. In their study, Nguyen et al. used mixtures of mAbs against HPAI H5/2014 2.3.4.4c and H5 LPAI to setup an immunochromatographic kit. The kit LoD was 10^4.8^–10^5.5^ EID50/test for H5 HPAI 2.3.4.4c and e subclades, and it could detect other H5Nx viruses, including 2.3.2.1c; H5 LPAI is not currently emergent nor prevalent worldwide. Additionally, the LoD for the HA specific target was lower (ten-fold) than that for the nucleoprotein conserved target in the detection of the same virus, and the kit could not distinguish between H5 LPAI and H5 HPAI viruses. The commercial H5-specific RDT kit commonly used in Korea (Catalog Number: RG1505MH, Bionote) also failed to detect H5 HPAIVs within the 2.3.4.4b subclade (Appendix A). In the present study, we developed two novel mAbs against the 2.3.4.4b subclade and conjugated these mAbs to Eu NPs, which are well-known fluorescence materials employed to improve the sensitivity of RDT kits [47,48,49,50]. Thus, our FICT assay obtained an LoD of 10^3.9^ EID50/test (Appendix A) for the detection of H5 2.3.4.4b viruses, which was ~7.9-fold higher than that reported by Nguyen et al. (2017), and 32-fold higher than that of the colloidal gold-based RDT kit using the same pair of mAbs. The LoD of our FICT was comparable with that of the commercial kit targeting the conserved region of influenza A H5 subtype (Catalog Number: RG1505MH, Bionote), with a 4-fold higher sensitivity (Appendix A). Additionally, our system was able to detect viruses in spiked samples of both chicken feces and human swabs, and to distinguish between H5 HPAI 2.3.4.4b and H5 LPAI viruses, as well as other AIVs subtypes.

In the case of influenza infection, feces samples are the most convenient to collect from poultry or during wild birds surveillance activities, because the highest viral load has been observed in chicken feces and these samples can be collected without disturbing the animal [22]. However, avian feces are acidic environments, and their high mass may cause antibody precipitation or bio-conjugate aggregation, thereby influencing RDT reactions and applicability in diagnostic systems [51]. In the present study, we optimized a lysis buffer containing 0.1 M Tris, 0.1 M EDTA, 0.5% Triton X-100, and 0.6% SDS (pH 9.0) for keeping LoD stability in the FICT when increasing the amount of feces samples.

Subclade 2.3.4.4 H5 HPAIVs show high virulence and severe lethality in chicken within 2 DPI [20,52]. In the present study, the A(H5N6)/Duck/Foshan/41/2019 isolate showed severe clinical signs, and all infected chicken died at 3 DPI; as such, sample collection and the RDT should be performed early, before animal death. Our FICT system showed 100% (4/4) sensitivity for early detection of H5N6 in stool samples at 2 DPI. However, a positive signal was not detected in the three cloacal samples collected from the same H5N6-infected group. This result corroborated the lower viral load of cloacal samples compared to fecal samples reported for AIVs [53,54], as this trend was observed for the H5N6 and H9N2 viruses included in the present study. We also observed more mucus in cloacal than in fecal samples, and thus our system should be further optimized to detect viruses in cloacal samples.

Real time-PCR is the gold standard method for detecting viral genes owing to its high sensitivity and specificity. A LoD of 5 × 10^1^ cDNA copies for the detection of H5 2.3.4.4 subclades by RT-PCR has been reported, but the viral loads of influenza in stool samples range between 10^4^–10^5^ cDNA copies per reaction [32]. In the present study, ~4.784 × 10^4^ cDNA copies for RT-PCR corresponded to the LoD of our H5 2.3.4.4c RGV assay using spiked stool samples. Thus, our system could detect H5 HPAIVs in stool samples.

In conclusion, we successfully developed novel H5 subtype-specific mAbs based on RGV and applied them to detect early and discriminate between H5 HPAIVs 2.3.4.4b and others in fecal samples using a rapid fluorescent diagnostic system with high sensitivity and specificity. This system may become an innovative diagnostic tool for the identification of H5 HPAIVs on site during outbreaks or surveillance activities.

## 4. Materials and Methods

### 4.1. Reagents

Europium (Eu) beads (1% w/t, 0.2 µm, Catalog Number: FCEU002) were obtained from Bangs Laboratories, Inc. (Fishers, IN, USA). N-(3-Dimethylaminopropyl)-N′-ethylcarbodiimide hydrochloride (EDC, Catalog Number: 22980-5G) and N-hydroxysulfosuccinimide sodium salt (Sulfo-NHS, Catalog Number: 56485-1G) were obtained from Sigma-Aldrich (St. Louis, MI, USA). The pHW200 vector was provided by Dr. Chris MOK, The University of Hong Kong.

### 4.2. Cell and Virus Cultures

Madin–Darby canine kidney cells (MDCK) (ATCC, Manassas, VA, USA) and 293T cells (provided by Dr. Chris MOK, The University of Hong Kong) were maintained in Eagle’s minimum essential medium [MEM, 25 mM N-2-hydroxyethylpiperazine-N’-2-ethanesulfonic acid, 1% antibiotic-antimycotic (AA), and 10% fetal bovine serum (FBS)]. AA 100× and FBS were purchased from Gibco™ (Thermo Fisher Scientific, Waltham, MA, USA). The influenza B virus was obtained from the Korea Disease Control and Prevention Agency. All viruses were inoculated on 10-days-embryonated chicken eggs or MDCK cells, and then titrated by HA assay, as previously described [55]. The viruses used in the present study are listed in Appendix A.

### 4.3. Generation of H5 HPAIVs Based on Reverse Genetics

The HA sequences of H5N6 clade 2.3.4.4b (A/Anas platyrhynchos/Korea/W612/2017) and H5N8 clade 2.3.4.4c (A/Mallard Duck/Korea/W452/2014) were provided by the Chungbuk National University College of Medicine. The polybasic cleavage site of each HA gene segment was deleted by gene mutagenesis to decrease virulence. The HA gene segments were then obtained using H5N6 sense (5’-GGCTCAGAAATAGTCCTCTAAGAGAAAAAAGAGGGCTGTTTGG-3’) and antisense (5’-CCAAACAGCCCTCTTTTTTCTCTTAGAGGACTATTTCTGAGCC-3’) primers and (H5N8) Sense primer: 5’-GGCTCAGAAATAGTACTCTAAGAGAAAGAAGAGGACTATTTGG-3’ antisense primer 5’-CCAAATAGTCCTCTTCTTTCTCTTAGAGTACTATTTCTGAGCC-3’. Seven internal gene segments were obtained from the PR8/H1N1 influenza A virus by polymerase chain reaction (PCR) using universal primer [37]. All gene segments obtained were cloned into the pHW2000 vector and H5 HPIAVs were reconstituted by the transfection of plasmids into a mixture of 293T and MDCK cells.

### 4.4. Construction and Expression of the HA Globular Head Protein Domain

Plasmids containing the H5N6-2.3.4.4b HA DNA sequence were used for PCR amplification of the globular head protein domain using the forward and reverse primers 5′-AAGCTTAAAACACACAACGGGAAGCTCTGCG-3′ and 5′-CTCGAGTCCAGAGCCACCACTAGAGTTTATCGCT-3′, respectively. The amplified PCR product was digested by *Hind*III and *Xho*I endonucleases and ligated into the pET21b+ vector for protein expression. The ligated plasmid was transformed into *E. coli* BL21 (DE3) competent cells, and protein expression was achieved as described previously [43]. The recombinant protein was purified using the HisPur™ Ni-NTA resin (Sigma-Aldrich) according to the manufacturer’s instructions.

### 4.5. Mouse Immunization and Hybridoma Preparation

Eight-week-old female BALB/c mice (Orient Bio Inc., Seongnam-si, Korea) were immunized three times with alive reverse genetic virus (RGV) H5 or H5N6 HA antigen produced by *E. coli* by footpad injection on hindfeet. Briefly, H5N6 HA antigen (50 µg/mouse) or H5 RGV [128 hemagglutination units (HAU)/mouse] were mixed with Freund’s incomplete adjuvant (Catalog Number: F5506, Sigma-Aldrich) or 100 µL Freund’s complete adjuvant (Catalog Number: F5881, Sigma-Aldrich), respectively, and injected on mice footpads. The clonal antibodies were obtained as previously described [29]. Briefly, mice sera were collected two weeks after each immunization and the antibody titer was checked by indirect enzyme-linked immunosorbent assay (ELISA). Two weeks after the last immunization, splenocytes and cells from lymphoid nodes were isolated, fused with a myeloma cell (Catalog Number: CRL1646, ATCC) at the ratio of 10:1 to 5:1 in the presence of polyethylene glycol, and transferred to a 96-well culture plate. After 16 h of incubation at 37 °C and 5% CO_2_, the hypoxanthine, aminopterin, and thymidine (HAT) medium was added. The HAT medium was changed daily. After three days, the hypoxanthine and thymidine (HT) medium was added, and the culture was maintained in the same incubation conditions for two weeks. When colonies appeared and the culture medium color changed to orange inside each well, cell supernatants were collected for ELISA to select the positive clones.

### 4.6. Indirect ELISA Assay

Viruses (H5, H1N1, H7N7) and chicken uninfected eggs fluid were diluted in coating buffer (bicarbonate/carbonate 100 mM, pH 9.6) to obtain the titer at 1000 HAU/mL. Then, 100 μL of each virus was coated on a 96-well microtiter plate (Greiner GmbH, Pleidelsheim, Germany) and incubated at 4 °C, overnight. After incubation, 200 μL phosphate buffered saline (PBS) plus 0.1% Tween 20 (PBS-T, pH 7.4) was used for washing. Following 5% non-fat dry milk blocking at 37 °C for 2 h, 100 μL candidate antibody (20 µg/mL) or cell supernatant was added to each well. After incubation for 1 h at 37 °C, the 96-well plate was washed with PBS-T before adding 100 μL horse-radish peroxidase (HRP)-conjugated rabbit anti-mouse IgG (Catalog Number: ab97046, Abcam, Cambridge, UK; 0.02 μg/well) to detect antigens. After washing the plate five times with PBS-T (pH 7.4) to remove nonspecific binding, 100 μL 3,3′,5,5′-tetra methyl benzamine substrate solution (Invitrogen, Carlsbad, CA, USA) was added for color development. The sample was covered with foil for 10 min and the reaction was stopped by adding 100 μL 0.18 M sulfuric acid to each well. Optical density (OD) was determined on a microplate reader at 450 nm (SpectraMax® M Series Multi-Mode Microplate, San Jose, CA 95134, USA).

### 4.7. Sandwich Fluorescence-Linked Immunosorbent Assay (FLISA)

Sandwich FLISA was performed as described previously [29]. Briefly, a black 96-well microtiter plate (Greiner GmbH) was coated with 100 µL/well of 10 µg/mL antibody candidate in coating buffer (pH 9.6) and incubated at 4 °C for 12 h. After 5% non-fat dry milk blocking at 37 °C for 2 h, the recombinant virus (100 µL/well) was added and the plate was incubated for another hour at 37 °C. After washing the plate with 200 μL PBS-T (pH 7.4), Eu nanoparticle (NP)-conjugated antibody (100 μL/well) was added and the plate was incubated at 37 °C for another hour for antigen detection. Stringent washing with PBS-T (pH 7.4) was performed five times to remove nonspecific binding, and 100 μL PBS was added to each well. Fluorescence was then measured in the Infinite F200 microplate reader (TECAN, Männedorf, Switzerland) at 355 nm (excitation) and 612 nm (emission) wavelengths.

### 4.8. Western Blot Analysis

Recombinant viruses (1000 HAU/well) or bovine serum albumin (BSA, 10 μg/well), used as the negative control, were subject to 10% sodium dodecyl sulphate–polyacrylamide gel electrophoresis (SDS-PAGE) at 100 V for 2 h. The gel was then soaked into a transfer buffer and proteins were bound onto a polyvinylidene difluoride (PVDF) membrane. The PVDF membrane was blocked with 5% non-fast milk for 2 h at 37 °C, diluted antibodies (10 μg/mL each) were added, and the membrane was incubated for 1 h at 37 °C. HRP-conjugated antibody (Abcam) was used for detection according to the manufacturer’s protocol. Finally, the substrate (Catalog Number: 1705060, Bio-Rad Laboratories, Inc., Hercules, CA, USA) was added for visualization of the band target using the ChemiDoc MP System (Catalog Number:1708280, Bio-Rad Laboratories, Inc.).

### 4.9. Immunofluorescence Assay (IFA)

The IFA was performed as previously reported [29]. First, MDCK cells were coated on glass coverslips and incubated at 37 °C and 5% CO_2_ overnight. After rinsing coverslips twice, recombinant viruses diluted in Dulbecco’s modified Eagle medium containing 1% antibiotic, 0.3% BSA, and 1 μg/mL trypsin from bovine pancreas were used to infect cells at multiplicity of infection (MOI) = 0.5 during a 12-h incubation at 37 °C and 5% CO_2_. Cells were fixed using 4% paraformaldehyde in PBS (pH 6.9) for 15 mins. After washing thrice with PBS-T (pH 7.4), 0.1% Triton X100 was added to each coverslip for permeabilization. Blocking was achieved using 5% BSA in PBS-T with 22.52 mg/mL glycine for 2 h at 37 °C. Primary and secondary antibodies were diluted at 1:3000 in blocking buffer and then incubated for 1 h and for 45 min, respectively. Finally, counter staining was performed by dropping 20 μL 4′,6-diamidino-2-phenylindole-mounting medium on a slide, inverting the coverslip on the slide for 15 min, and then sealing the coverslip with nail polish to prevent the drying and movement of cells under the microscope.

### 4.10. Measurement of Binding Affinities

Surface plasmon resonance (SPR) was performed to characterized mAbs or analyze binding affinities between antibodies and H5N6 HA antigen. The methods are described in the Method section of the Appendix A.

### 4.11. Conjugation of Eu NP and Gold (Au) NP Antibodies

The Eu NP-conjugated antibody #11.4 was prepared through a well-established covalent procedure (Bangs Laboratories Inc.). Briefly, 20 µL Eu (1% w/t, 0.2 µm) was added to 645 µL 0.1 M Tris-HCl (pH 7.0) in the presence of EDC and Sulfo-NHS (at final concentrations of 0.13 mM and 10 mM, respectively) and allowed to react for 1 h at 37 °C. The surplus of EDC and Sulfo-NHS was removed via centrifugation at 24,000× *g* for 5 min. The activated Eu was mixed with 60 µL of antibody (1 mg/mL) in 940 µL 0.1 M sodium phosphate (pH 8.0) and reacted for 2 h at 30 °C. After centrifugation at 24,000× *g* for 5 min, the collected bio conjugates were washed with storage buffer (2 mM borax pH 9.0 plus 0.1% BSA pH 8.0). Finally, the Eu NP-conjugated antibody was re-suspended in 100 µL storage buffer and kept at 4 °C.

Gold nanoparticles (Au NPs) were purchased from Bore Da Biotech, Korea (Catalog Number: BBCG.40) and then conjugated to mAbs through the adsorption method. Briefly, 60 μL Au NPs (OD = 3) and 50 μL mAbs (1 mg/mL) were added into 940 μL 20 mM Tris base buffer (pH 9.5). The mixture was incubated for 3 h with gentle rotation at 25 °C and then centrifuged at 24,000× *g* for 10 min to remove the supernatant. The final product was washed and preserved in 200 μL storage buffer (100 μL 2 mM borax pH 9.0 plus 0.1% BSA pH 8.0). Then, 6 μL of the stock solution was dropped into each conjugate pad and dried for 20 min at 30 °C.

### 4.12. Lateral Flow Test Strips for Fluorescent Immunochromatography

Each test strip consisted of four components: a sample application pad, a conjugate pad, a nitrocellulose membrane, and an absorbent pad. The test line (TL) of each strip was prepared by dispensing 4 mg/mL mouse anti-influenza H5 subtype-specific mAb #23.3 and the control line (CL) by dispensing 0.5 mg/mL IgG mouse. Strips were dried at 30 °C for two days. Eu NP-conjugated or Au NPs-conjugated mAb #11.4 (6 µL each) was dropped onto the conjugate pad for use as the detector antibody, and the pad was dried at 30 °C for 15 mins. Then, 75 µL of sample and 125 µL of lysis buffer were dropped onto the sample pad and incubated for 15 mins to complete the reaction. The results of test strips were read in a Medisensor Inc. (Daegu, Korea) fluorescence reader device at 355 nm (excitation) and 612 nm (emission) wavelengths.

### 4.13. Spiked Sample Preparation

A two-fold dilution of H5/2.3.4.4b RGV was prepared in deionized sterile water and human nasal swab. In case of spiked mimic samples of chicken stool, 0.3 g of chicken stool was mixed with 0.3 mL of diluted H5/2.3.4.4b RGV and then transferred to 0.5 mL of lysis buffer by using cotton swabs. Finally, 200 µL of each sample was dropped onto a test strip after a 30 s spin. The same volume of sample was used for FICT and commercial RDT kit assays and for RNA extraction for RT-PCR. The limit of detection (LoD) of the virus titer was determined by the limit of blank (LOB), as described previously [56].

### 4.14. Reverse Transcription PCR (RT-PCR)

The RT-PCR was performed as described previously [29]. Briefly, viral RNA was obtained by using the viral RNA extraction kit (QIAGEN, Hilden, Germany). The Quantitect Probe RT-PCR kit (QIAGEN) was used for preparing the RT-PCR reaction using the H5 primers, probes, and conditions described in a previous report [57]. For obtaining a standard RNA copy number, the RNA template was ligated into the pGEM-T Easy (Promega, Madison, WI, USA) plasmid including a 161-base pair (bp) HA insert. For the in vitro transcription of HA RNA, the RiboMax kit (Promega) was used to determine the RNA copy number necessary for the LoD of FICT. A standard curve was calculated automatically by plotting the cycle threshold (Ct) values against each standard with known RNA copy number and by extrapolating the linear regression of this curve. The PCR products were analyzed in agarose gels (2%) for 90 mins at 35 volts.

### 4.15. Evaluation of RDT Performance

To evaluate the performance of our FICT, its LoD was compared to that of the commercial Influenza Ag test for detecting H5 Influenza virus type A (Catalog Number: RG1505MH, Bionote, Hwaseong-si, Korea) by following the manufacturer’s instructions. The SD BIOLINE Influenza Ag A/B/A(H1N1) RDT (Standard Diagnostics, Inc., Suwon-si, Korea) was also used for the normalization of virus quantification between the two systems.

### 4.16. Experimental Infection in the Chicken Model 

The experimental infection of chicken was performed in a biosafety level (BSL) 3 facility in the Zoonosis Research Institute at Jeonbuk National University (KZRI). Briefly, 4-week-old white-leghorn chicken were intranasally infected with 100 µL 10^5^ EID50 of A(H5N6)/Duck/Foshan/41/2019 and H9N2 virus (infected group) or PBS 1× (non-infected group) (*n* = 3 chicken/group). Fresh fecal samples were randomly collected two days after infection.

### 4.17. Statistical Analyses

Means and standard deviations (SDs) were calculated for each tested group and one- and two-way analysis of variance (ANOVA) were performed using GraphPad Prism 5.0 (GraphPad Software Inc., San Diego, CA, USA). *p* < 0.05 was considered significant between the control and testing of comparable groups.

## Figures and Tables

**Figure 1 ijms-23-06301-f001:**
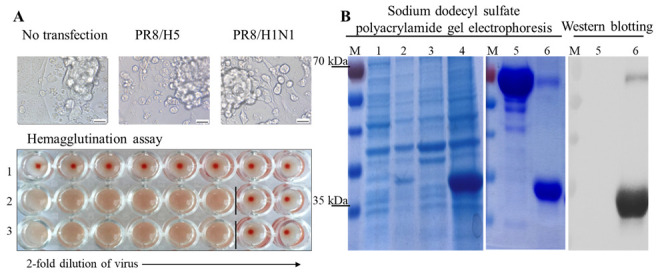
Preparation of recombinant virus and immunogenic protein. (**A**) Expression of recombinant influenza H5 hemagglutinin (HA) 2.3.4.4b virus based on the H1N1/PR8 virus backbone 3 days post-transfection. The left panel shows plasmids in the 293T and Madin–Darby canine kidney cells (MDCK) cells mixture (scale bar, 20 μm; original magnification, 400×), and the results of the hemagglutination assay performed for checking the titer of the recombinant virus without transfection (1), in PR8/H5 HA 2.3.4.4b (2), and in PR8/H1N1 (3). The virus titer was 1280 HAU/mL for both PR8/H1N1 and PR8/H5 2.3.4.4b (HA). (**B**) The right panel shows expression of the globular head domain H5N6 (HA) 2.3.4.4b protein in *E. coli*. Sodium dodecyl sulfate–polyacrylamide gel electrophoresis (SDS-PAGE) and Western blotting results are shown, M: maker; 1: non-induced lysis; 2: induced lysis; 3: non-induced pellet; 4: induced pellet; 5: BSA; and 6: final product after purification of the globular head domain protein (20 µg/lane).

**Figure 2 ijms-23-06301-f002:**
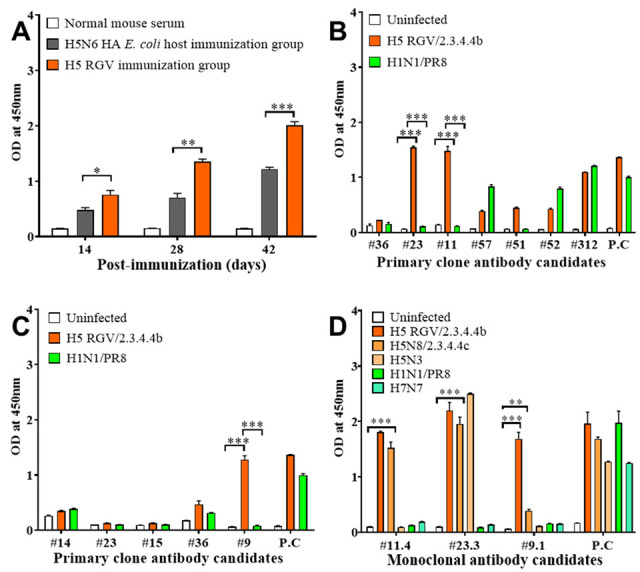
Development of H5-specific antibodies. Mice were immunized with H5 reversed genetic influenza virus (RGV) or recombinant protein three times by footpad injection. (**A**) Mice sera were collected 2 weeks after each immunization for determining the levels of antibody against H5 RGV and other subtypes as determined by indirect enzyme-linked immunosorbent assay (ELISA) in the experimental and control groups. (**B**) Levels of primary clone antibodies for H5 RGV. (**C**) Levels of primary clone antibodies for H5N6 HA recombinant antigen. (**D**) Monoclonal antibodies against several H5 subtypes, H1N1/PR8, and H7N7. The positive control (P.C) is the anti-NP of influenza A virus. ANOVA was used to analyze significant differences between the experimental groups and the control group. *, *p* < 0.05; **, *p* < 0.01; ***, *p* < 0.001.

**Figure 3 ijms-23-06301-f003:**
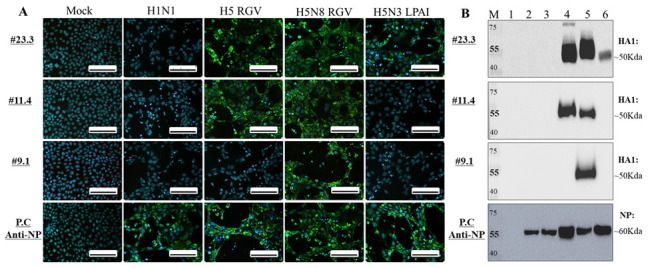
Characterization of H5-specific mAbs by immunoassay. (**A**) Immunofluorescence assay (IFA). Cells were coated and fixed on a coverslip using 4% paraformaldehyde, infected for 12 h with H1 and H5 subtypes at multiplicity of infection (MOI) = 0.5, and then incubated with the mAb candidates #23.3, #11.4, and #9.1. After washing, fluorescence was detected by a FITC-conjugated secondary antibody. Scale bar, 100 μm. Original magnification, 400×. Mock, no virus infection. (**B**) Western blot assay; 500 HAU of each virus was loaded per lane and 10% SDS-PAGE was performed for 2 h. The proteins were then transferred onto a polyvinylidene difluoride (PVDF) membrane for 3 h. After 5% non-fat milk blocking, the membrane was incubated with primary mAb candidates and then with the secondary antibody HRP. M, marker; 1, BSA; 2, H1N1; 3, H9N2; 4, H5/2.3.4.4b RGV; 5, H5N8/2.3.4.4c RGV; 6, H5N3 LPAI. The positive control (P.C) is the anti-NP of influenza A virus.

**Figure 4 ijms-23-06301-f004:**
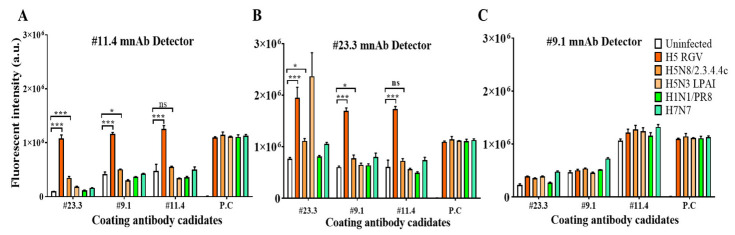
Selection of a specific pair of mAb candidates for H5 HPAIVs detection by sandwich FLISA. The selected mAbs were coated on 96-well plates as capture elements and then detect by Eu NP-conjugated antibodies. (**A**). #11.4 mAb detector, (**B**). #23.3 mAb detector, (**C**). #9.1 mAb detector. The positive control (P.C) is the anti-NP of influenza A virus. One-way ANOVA was used to analyze the data. * *p* < 0.05; *** *p* < 0.001; ns, not significant.

**Figure 5 ijms-23-06301-f005:**
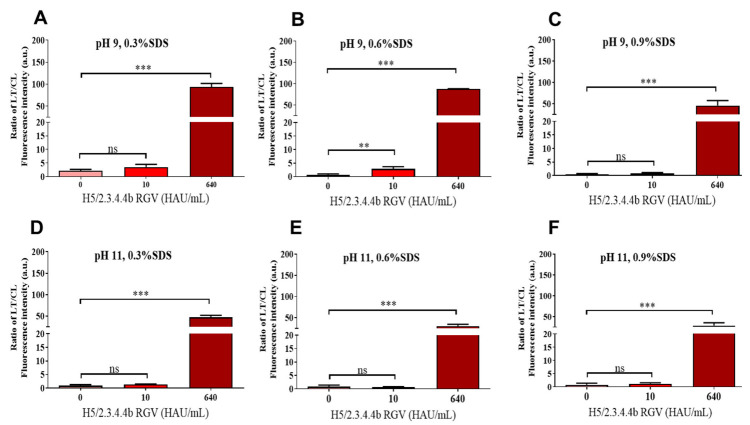
Optimization of lysis buffer for FICT of the Eu bio-conjugate #11-4 and #23.3 strip coating. Various concentrations of SDS (0.3, 0.6, and 0.9%) at pH 6.0 (Appendix A), 9.0 (**A**–**C**), and 11.0 (**D**–**F**) were tested in basic lysis buffer (0.1 M Tris, 0.1 M EDTA, and 0.5% Triton X-100). One-way ANOVA was used to analyze the FICT data. ** *p* < 0.01; *** *p* < 0.001; ns, not significant. Raw fluorescence peaks from the test line (TL) and control line (CL) in FICT are shown in Appendix A.

**Figure 6 ijms-23-06301-f006:**
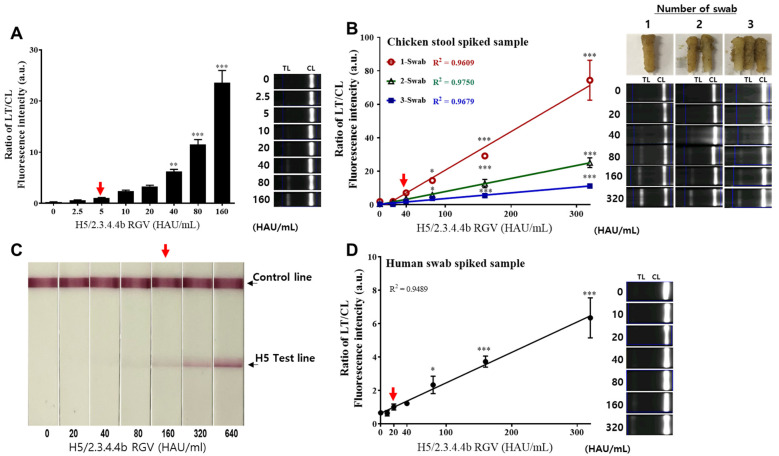
FICT LoD against serial dilution of H5 RGV. Serial dilution of H5/2.3.4.4b RGV was performed in deionized sterile water (DW) or mimic samples and then subject to FICT using a colloidal gold-based RDT kit. (**A**). TL/CL ratio in DW. (**B**). TL/CL ratio in chicken stool spiked samples. 0.3 g of chicken stool was mixed with 0.3 mL of diluted H5/2.3.4.4b RGV and then transferred to 0.5 mL of lysis buffer by using cotton swabs. Finally, 200 µL of each sample was dropped onto a test strip after a 30 s spin. (**C**). Colloidal gold-based RDT kit with the selected mAb pair. (**D**). Spiked human nasal swab samples. After mixing the diluted virus in the human swab sample, 75 µL of the sample plus 125 µL of lysis buffer was used for FICT. The red arrow indicates the virus titer representing the LoD of FICT. Data are mean ± SD (*n* = 3). The linear regression (dotted line) showed R^2^ > 0.9. One-way ANOVA was used to analyze the FICT data. * *p* < 0.05; ** *p* < 0.01; *** *p* < 0.001 in relation to the control group. Raw fluorescence peaks for the test line (TL) and control line (CL) of FICT are shown in Appendix A.

**Figure 7 ijms-23-06301-f007:**
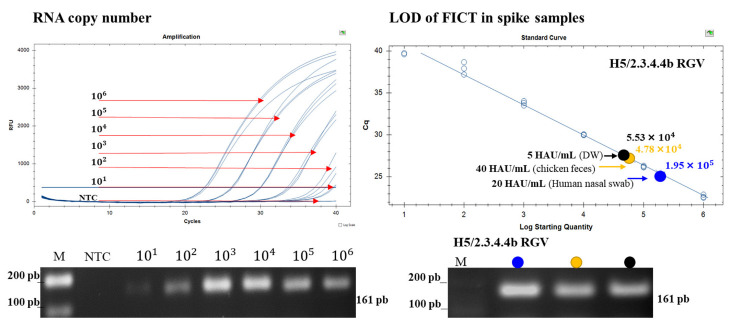
Assessment of FICT performance by RT-PCR. The linear relationship between Ct and RNA copy number is shown. The bottom panel shows the PCR products of different RNA standards and diluted virus for LoD determination (5 µL sample/lane); target bands were obtained at 161 base pairs (bp). The red arrows indicate the point corresponding to the RNA copy number and the color dots correspond to the LoD of the virus titer in the different samples. NTC, no template control. Raw RT-PCR data are shown in Appendix A.

**Figure 8 ijms-23-06301-f008:**
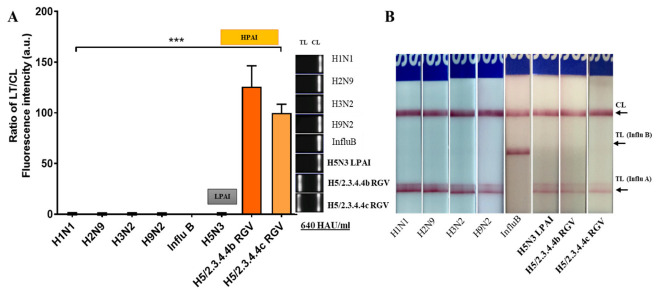
Specificity of the FICT assay. (**A**). TL/CL ratio in the FICT assay. Data are mean ± SD (n = 3) for other subtypes of virus at high titer (640 HAU/mL). (**B**). SD RDT commercial kit for normalization of virus amount. Raw fluorescence peaks of the test line (TL) and control line (CL) in FICT are shown in Appendix A.

**Figure 9 ijms-23-06301-f009:**
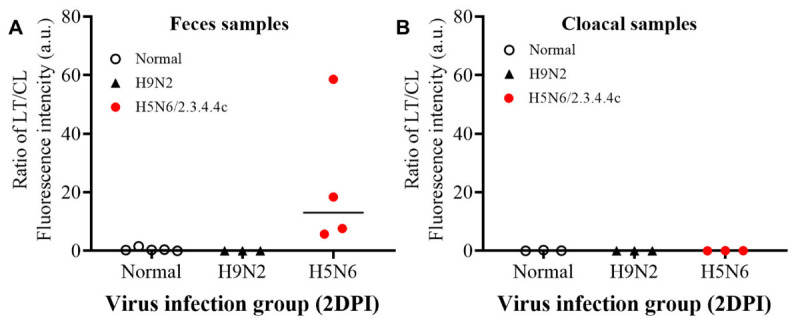
Performance of FICT for H5 HPAIVs detection in fecal samples. Viruses were inoculated into 4-weeks-old specific-pathogen-free chicken. Fresh stool and cloacal samples were collected 2 days post infection (2DPI) and then subjected to FICT, Au NPs-RDT, and RT-PCR assays to determine the sensitivity of diagnostic tests. (**A**) TL/CL ratio in fecal samples. (**B**) TL/CL ratio in cloacal samples. H5N6 HPAI and H9N2 infection experiments were performed in the BSL3 and BLS2 laboratory, respectively. Raw data are shown in Appendix A.

**Figure 10 ijms-23-06301-f010:**
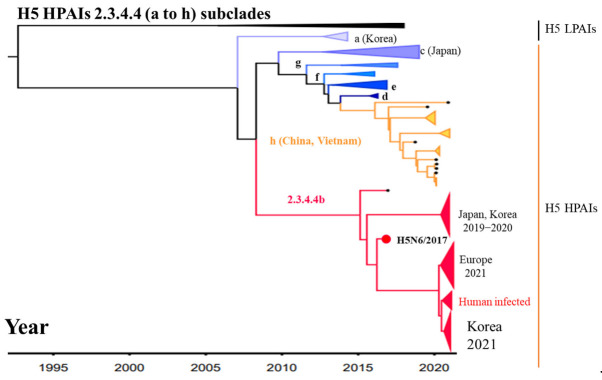
Distribution of the H5 HPAIVs 2.3.4.4 subclades (a to h) in Korea and other world regions in 2014–2021. A Bayesian time-resolved phylogenetic tree was obtained for the hemagglutinin gene segment using BEAST 1.8.4. Isolate sequences were downloaded from the Global Initiative on Sharing Avian Influenza Data database (http://www.gisaid.org; accessed on 25 December 2021). the red circle indicates the A/Anas platyrhynchos/Korea/W612/2017(H5N6) 2.3.4.4b gene section used as the immunogenic antigen in the present study.

**Table 1 ijms-23-06301-t001:** Comparison of FICT, colloidal gold-RDT, and RT-PCR assays diagnostic performance on clinical samples.

Clinical Specimens	Virus Infection Group	Sensitivity (%)		
		2 Days Post Infection		
		RT-PCR Test (Positive/Negative)	FICT Assay (Positive/Negative)	Au NPs-RDT (Positive/Negative)
Feces samples	Normal ^a^	(0/5)	0% (0/5)	0% (0/5)
	H9N2 ^b^	(2/3)	0% (0/3)	0% (0/3)
	H5N6 ^c^	(4/4)	100% (4/4)	50% (2/4)
Cloacal samples	Normal	(0/3)	0% (0/3)	0% (0/3)
	H9N2	(3/3)	0% (0/3)	0% (0/3)
	H5N6	(2/3)	0% (0/3)	0% (0/3)

^a^ Non-infected group. ^b^ A/Chicken/Korea/LPM429/2016 (H9N2). ^c^ A(H5N6)/Duck/Foshan/41/2019) 2.3.4.4c.

## Data Availability

The data presented in this study are available in Appendix A.

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
