# Peer review of "Development of a Rapid Fluorescent Diagnostic System for Early Detection of the Highly Pathogenic Avian Influenza H5 Clade 2.3.4.4 Viruses in Chicken Stool"

_ijms, 2022, doi:10.3390/ijms23116301_

Round 1
Reviewer 1 Report
This manuscript describes the results of development and sensitivity/specificity assessment of a rapid antigenic test for diagnosis of H5Nx avian influenza virus in chicken stool. The authors not only optimized the pair of capture/detecting monoclonal antibodies, but also found an optimal lysis buffer for sample processing. The paper presents all steps for the development of the test system, starting from antigen preparation and finishing by its testing on the infected SPF chicken. The manuscript and the Supplementary materials contain all the details that support the authors’ conclusions. The study is important and the results can be used by poultry producers for rapid diagnosis of possible H5Nx infections. The strength of the developed test system is that the virus can be detected in stool, and there is no need to deal with animals for sample collection. In general, the manuscript is well written, but there are some suggestions that may help improve it before publication.
Major
The authors should be more clear about the goal of developing a test system: do they want to make a system sensitive to only one specific lineage of the H5 viruses in order to distinguish it from other H5 viruses, or do they want to develop a test for the detection of all the most important H5 lineages?
Minor
· Clinical studies mean that something is being tested in humans. Please correct the title of the study of the test system in chickens.
· The resolution of some figures could be improved for better reading the details. Possibly, the settings of the Word processor can help (Word Options -> Advanced -> Image Size and Quality -> Default resolution -> Do Not Compress Images).
· There are multiple grammar errors throughout the manuscript, please check in carefully.
· The abbreviation mAb is more commonly used than mnAb.
Author Response
Comments and Suggestions for Authors
This manuscript describes the results of development and sensitivity/specificity assessment of a rapid antigenic test for diagnosis of H5Nx avian influenza virus in chicken stool. The authors not only optimized the pair of capture/detecting monoclonal antibodies, but also found an optimal lysis buffer for sample processing. The paper presents all steps for the development of the test system, starting from antigen preparation and finishing by its testing on the infected SPF chicken. The manuscript and the Supplementary materials contain all the details that support the authors’ conclusions. The study is important and the results can be used by poultry producers for rapid diagnosis of possible H5Nx infections. The strength of the developed test system is that the virus can be detected in stool, and there is no need to deal with animals for sample collection. In general, the manuscript is well written, but there are some suggestions that may help improve it before publication.
First of all, we would like to thank the reviewer for your valuable and detailed comments. It is verry helpful for improvement of our manuscript qualities.
Major
The authors should be more clear about the goal of developing a test system: do they want to make a system sensitive to only one specific lineage of the H5 viruses in order to distinguish it from other H5 viruses, or do they want to develop a test for the detection of all the most important H5 lineages?
Response: The diagnostic test system is very great if it is able to detect all the important and emerging variants of H5 HLPA lineages specifically. However, it seems to be hardly possible to make those system using specific monoclonal antibodies, because of the genetic variation even within H5 highly pathogenic avian virus (HPAIVs) lineages. Therefore, we aimed to develop a test system for H5Nx subclade 2.3.4.4, one of the most emerging variant subclades of the H5 HPAIVs lineage, in order to distinguish it from other H5 low pathogenicity avian influenza viruses [1, 2].
Revision lines: 81-84, “In the present study, we aimed to development of a rapid fluorescent diagnostic system for early detection of the H5 HPAIVs subclade 2.3.4.4, one of the most emerging variant subclades of the H5 HPAIVs lineage, in order to distinguish it from other H5 LPAIVs in chicken feces”
Minor
- Clinical studies mean that something is being tested in humans. Please correct the title of the study of the test system in chickens.
Response: Thank review for your insightful comment. We have corrected the error.
Revision lines: 446, “Evaluation of FICT system in the chicken model”
- The resolution of some figures could be improved for better reading the details. Possibly, the settings of the Word processor can help (Word Options -> Advanced -> Image Size and Quality -> Default resolution -> Do Not Compress Images).
Response: Thanks for your detailed guidance. Flowing your suggestion, we have changed and updated new figures.
Revision figures: 2, 3, 4, 5, 6.
- There are multiple grammar errors throughout the manuscript, please check in carefully.
Response: Dear reviewer, according to your comment, we have sent this manuscripts version to proofreading service for grammatical, spelling and punctuation errors checking. We attached a certificate from Editage, one of professional English Proofreading Services.
- The abbreviation mAb is more commonly used than mnAb.
Response: We have changed all abbreviation mnAb to mAb.
Revision lines: 22, 24, 33, 85, 88, 205, 220, 221, 230, 232, 292, 306, 523, 573.
Reference
- Duong, B.T., et al., Assessing Potential Pathogenicity of Novel Highly Pathogenic Avian Influenza (H5N6) Viruses Isolated from Mongolian Wild Duck Feces Using a Mouse Model. Emerg Microbes Infect, 2022: p. 1-29.
- Lewis, N.S., et al., Emergence and spread of novel H5N8, H5N5 and H5N1 clade 2.3.4.4 highly pathogenic avian influenza in 2020. Emerg Microbes Infect, 2021. 10(1): p. 148-151.
Reviewer 2 Report
This manuscript by Bao Tuan Duong and co-workers, entitled “Development of a Rapid Fluorescent Diagnostic System for Early Detection of the Highly Pathogenic Avian Influenza H5 clade 2.3.4.4b viruses in Chicken Stool”, addresses the development of two novel specific monoclonal antibodies for clade 2.3.4.4 H5Nx viruses to be used for developing a fluorescent immunochromatographic rapid strip test for the early detection of H5Nx influenza viruses on chicken stools samples. The development of high-performance H5 Highly Pathogenic Avian Influenza virus rapid diagnostic tests for fecal samples will be key for the identification and surveillance of viruses in poultry, wild birds, and human. Results obtained showed that mammalian reverse genetics expression system is a more advantageous technique for developing H5Nx- specific monoclonal antibodies than the E. coli expression host.
The purpose of the research is important in that viruses causing poultry outbreaks, in particular A(H5N6) 2.3.4.4 viruses, can also cause human H5 Highly Pathogenic Avian Influenza (HPAI) cases, also resulting in dead. So diagnostic systems targeting 2.3.4.4 subclades of HPAI H5 viruses could prevent the spreading of HPAI as a primary screening clinical tool.
The paper is well written and understandable to a specialist readership. The title clearly indicates the focus of the paper and the Abstract section satisfactorily summarizes the contents of the article. In the “Introduction” the context of the subject area is adequately addressed to justify the study and the objective of the manuscript is clearly stated. "Materials and methods" are suitable, containing well-defined information, appropriate to the project and sufficient for understanding and replication of the research. Figures, Tables and supplementary materials are well designed and all necessary for understanding of the text.
The research group has been involved in the subject for a long time and this paper represents the continuation of research already carried out successfully.
I have just a few suggestions.
Abstract:
First line, please delete therefore.
Please change LOD with LoD, or change loD with LOD throughout the text.
References:
References must be rewritten in accordance with the journal.
I would suggest adding other papers, such as those listed below, to make the discussion even more complete.
Yeo SJ, Huong DT, Hong NN, Li CY, Choi K, Yu K, Choi DY, Chong CK, Choi HS, Mallik SK, Kim HS, Sung HW, Park H. Rapid and quantitative detection of zoonotic influenza A virus infection utilizing coumarin-derived dendrimer-based fluorescent immunochromatographic strip test (FICT). Theranostics. 2014 Sep 25;4(12):1239-49. doi: 10.7150/thno.10255. PMID: 25285172; PMCID: PMC4184001.
Yu ST, Thi Bui C, Kim DTH, V T Nguyen A, Thi Trinh TT, Yeo SJ. Clinical evaluation of rapid fluorescent diagnostic immunochromatographic test for influenza A virus (H1N1). Sci Rep. 2018 Sep 7;8(1):13468. doi: 10.1038/s41598-018-31786-8. PMID: 30194330; PMCID: PMC6128899.
Zhang P, Vemula SV, Zhao J, et al. A highly sensitive europium nanoparticle-based immunoassay for detection of influenza A/B virus antigen in clinical specimens. J Clin Microbiol. 2014;52(12):4385-4387. doi:10.1128/JCM.02635-14
Supplementary:
Figure S11
Legend: please change “form” with “from”.
Author Response
Comments and Suggestions for Authors
This manuscript by Bao Tuan Duong and co-workers, entitled “Development of a Rapid Fluorescent Diagnostic System for Early Detection of the Highly Pathogenic Avian Influenza H5 clade 2.3.4.4b viruses in Chicken Stool”, addresses the development of two novel specific monoclonal antibodies for clade 2.3.4.4 H5Nx viruses to be used for developing a fluorescent immunochromatographic rapid strip test for the early detection of H5Nx influenza viruses on chicken stools samples. The development of high-performance H5 Highly Pathogenic Avian Influenza virus rapid diagnostic tests for fecal samples will be key for the identification and surveillance of viruses in poultry, wild birds, and human. Results obtained showed that mammalian reverse genetics expression system is a more advantageous technique for developing H5Nx- specific monoclonal antibodies than the E. coli expression host.
The purpose of the research is important in that viruses causing poultry outbreaks, in particular A(H5N6) 2.3.4.4 viruses, can also cause human H5 Highly Pathogenic Avian Influenza (HPAI) cases, also resulting in dead. So diagnostic systems targeting 2.3.4.4 subclades of HPAI H5 viruses could prevent the spreading of HPAI as a primary screening clinical tool.
The paper is well written and understandable to a specialist readership. The title clearly indicates the focus of the paper and the Abstract section satisfactorily summarizes the contents of the article. In the “Introduction” the context of the subject area is adequately addressed to justify the study and the objective of the manuscript is clearly stated. "Materials and methods" are suitable, containing well-defined information, appropriate to the project and sufficient for understanding and replication of the research. Figures, Tables and supplementary materials are well designed and all necessary for understanding of the text.
The research group has been involved in the subject for a long time and this paper represents the continuation of research already carried out successfully.
We are grateful to the reviewer for your detail comments and suggestions. Your comments is very helpful for improvement of our manuscript qualities.
I have just a few suggestions.
Abstract:
First line, please delete therefore.
Please change LOD with LoD, or change loD with LOD throughout the text.
Response: We have corrected all error by flowing your comment.
Revision lines: 20, “deleted therefore”; 27, “LOD with LoD”.
References:
References must be rewritten in accordance with the journal.
I would suggest adding other papers, such as those listed below, to make the discussion even more complete.
- Yeo SJ, Huong DT, Hong NN, Li CY, Choi K, Yu K, Choi DY, Chong CK, Choi HS, Mallik SK, Kim HS, Sung HW, Park H. Rapid and quantitative detection of zoonotic influenza A virus infection utilizing coumarin-derived dendrimer-based fluorescent immunochromatographic strip test (FICT). Theranostics. 2014 Sep 25;4(12):1239-49. doi: 10.7150/thno.10255. PMID: 25285172; PMCID: PMC4184001.
- Yu ST, Thi Bui C, Kim DTH, V T Nguyen A, Thi Trinh TT, Yeo SJ. Clinical evaluation of rapid fluorescent diagnostic immunochromatographic test for influenza A virus (H1N1). Sci Rep. 2018 Sep 7;8(1):13468. doi: 10.1038/s41598-018-31786-8. PMID: 30194330; PMCID: PMC6128899.
- Zhang P, Vemula SV, Zhao J, et al. A highly sensitive europium nanoparticle-based immunoassay for detection of influenza A/B virus antigen in clinical specimens. J Clin Microbiol. 2014;52(12):4385-4387. doi:10.1128/JCM.02635-14
Response: The references were re-check and rewritten flowing MDPI reference List and Citations Style Guide. As per your suggestion, we have also checked and added more included three above references in this discussion part.
Revision lines: 539, “50-53”
Supplementary:
Figure S11
Legend: please change “form” with “from”.
Response: We have corrected the error and uploaded revised supplementary data.
Revision lines: 140, “from”